# Integrated Genomic and Transcriptomic Elucidation of Flowering in Garlic

**DOI:** 10.3390/ijms232213876

**Published:** 2022-11-10

**Authors:** Einat Shemesh-Mayer, Adi Faigenboim, Tomer E. Ben Michael, Rina Kamenetsky-Goldstein

**Affiliations:** Institute of Plant Sciences, Agricultural Research Organization, The Volcani Institute, Rishon LeZion 7505101, Israel

**Keywords:** *Allium sativum*, florogenesis, flowering-key genes, PEBP family genes, *LEAFY*-like genes, gene duplication

## Abstract

Commercial cultivars of garlic are sterile, and therefore efficient breeding of this crop is impossible. Recent restoration of garlic fertility has opened new options for seed production and hybridization. Transcriptome catalogs were employed as a basis for garlic genetic studies, and in 2020 the huge genome of garlic was fully sequenced. We provide conjoint genomic and transcriptome analysis of the regulatory network in flowering garlic genotypes. The genome analysis revealed phosphatidylethanolamine-binding proteins (PEBP) and *LEAFY* (*LFY*) genes that were not found at the transcriptome level. Functions of *TFL*-like genes were reduced and replaced by *FT*-like homologs, whereas homologs of *MFT*-like genes were not found. The discovery of three sequences of *LFY*-like genes in the garlic genome and confirmation of their alternative splicing suggest their role in garlic florogenesis. It is not yet clear whether AsLFY1 acts alone as the “pioneer transcription factor” or AsLFY2 also provides these functions. The presence of several orthologs of flowering genes that differ in their expression and co-expression network advocates ongoing evolution in the garlic genome and diversification of gene functions. We propose that the process of fertility deprivation in garlic cultivars is based on the loss of transcriptional functions of the specific genes.

## 1. Introduction

The popular vegetable and nutraceutical crop garlic (*Allium sativum* L.) is widely used for culinary purposes, as well as in folk and conventional medicine. It is endorsed as a natural cure for numerous diseases. As a result, its popularity continues to grow. Increasing demand for garlic products with specific characteristics calls for its breeding and selection for new and desirable traits, and its adaptation to different climatic conditions. However, garlic breeding is complicated and challenging. Similar to potato, mango, *Curcuma*, and cassava [1], commercial garlic varieties do not produce seeds. They are only propagated vegetatively, and, therefore, garlic breeding depends on fertility restoration and the use of modern breeding tools [2].

Recent research has shown that garlic clones differ in their reproductive traits: flower initiation and bolting are either completely impaired, or small bulblets (topsets) that develop within the inflorescence compete with flowers for nutrients. Even if individual flowers of some genotypes succeed to differentiate, seed production is compromised due to male or female sterility, tapetum degeneration, and pollen abortion. However, a few genotypes have been found to produce viable reproductive organs and seeds [3,4,5,6].

*Allium* species have extremely large genomes [7], making them challenging to sequence. Garlic is the first *Allium* crop with a fully sequenced genome, which was published only recently [8]. Before that, PCR analysis and RNA-sequencing technologies were employed to generate genetic records [9,10,11]. In flowering garlic genotypes, morpho–physiological and transcriptome analyses have indicated that the first critical stage in the reproductive process is the induction and transition of the apical meristem from vegetative to reproductive stage when floral repressors are downregulated and floral inducers are expressed [12]. Garlic homologs of meristem identity genes, e.g., *LEAFY* (*LFY*), *APETALA1* (*AP1*), and *CAULIFLOWER* (*CAL*), have been identified, and their regulation during floral initiation recorded [12,13,14]. After the meristem transition, floral differentiation and scape elongation require a sugar supply [15,16], while unfavorable environmental conditions might cause male and female sterility in the developing flowers [17]. 

Similar to many other plant species, garlic development is significantly affected by regulation of genes of the phosphatidylethanolamine-binding protein (PEBP) family [12,15,18,19]. This family is conserved from prokaryotes to eukaryotes and is classified into three main groups: *MOTHER OF FT AND TFL1* (*MFT*)-like, *FLOWERING LOCUS T* (*FT*)-like, and *TERMINAL FLOWER1*/*CENTRORADIALIS (TFL1*/*CEN*)-like genes [20]. Plant evolution and domestication resulted in the duplication, divergence, and subfunctionalization of the PEBP family [18,20,21,22]. These genes integrate external and internal signals and pathways during the plant’s life cycle and have emerged as central regulators of plant development and flowering. Two groups, the *TFL1*-like and *FT*-like genes, have very high sequence similarity; however, in many plants, their functions are antagonistic: *FT* homologs act as flowering inducers, “florigens”, whereas *TFL1* homologs suppress the expression of floral identity genes by creating a repression complex with the transcription factor (TF) FD. *MFT*-like genes might be involved in flower development, although their main function is to control seed germination [23,24].

Subfunctionalization of *FT/TFL* genes has resulted in multiple orthologs and paralogs. For instance, in potato, floral and tuberization transitions are controlled by two different *FT*-like paralogs—*StSP3D* and *StSP6A*—that respond to independent environmental cues [25]. *FT1* homologs of *Brachypodium* and wheat respond dynamically to photoperiod extension and promote early inflorescence development [26,27]. In monocotyledonous geophytes, functionalization of PEBP genes results in multiple *FT* paralogs functioning in both sexual and vegetative reproduction [18]. Thus, *Lilium LlFT* is involved in meristem competence to environmental signals, whereas tulip *TgFT2* acts as a florigen. Another tulip PEBP gene, *TgFT3*, probably has bulb-specific functions [28]. In *Narcissus,* the *FT-like NtFT* is responsive to heat, and its transcript correlates with floral induction independently of photoperiod and vernalization [29,30]. Seven *FT-like* genes have been identified in the transcriptome of onion *(Allium cepa). AcFT1* and *AcFT4* functions and expression in the regulation of bulbing are antagonistic: *AcFT1* acts as a promoter and *AcFT4* as an inhibitor of bulbing, whereas *AcFT2* promotes flowering time [31]. Photoperiod, temperature, and drought greatly affect *FT* transcript levels in onion [32]. In the garlic transcriptome, six *FT*-like genes and one *TFL2*-like gene were found to be involved in bulb formation and flower induction [12]. Garlic *AsFT2* is upregulated upon vernalization and presumably acts as the garlic florigen, promoting florogenesis [12,15,16]. 

The current model of meristem development implies that during the vegetative phase, PEBP member TFL1 forms a complex with FD, which functions as an anti-florigen TF and represses the key flowering gene *LFY*. Following flower induction, the florigen *FT* is upregulated and competes with *TFL1* to prevent binding of the TFL–FD complex to the *LFY* locus [23,33,34]. Then, *LFY* binds to nucleosomes in closed chromatin regions and opens the chromatin by displacing linked histones and recruiting the chromatin-remodeling complex. These changes allow the binding of other TFs, leading to expression of the floral meristem identity genes. Due to these unique traits, LFY is perceived as a “pioneer TF in plants”, with high competence in shaping the plant’s epigenetic landscape [35]. 

Previous studies by our research group have found that *LFY* homologs play a major role in garlic florogenesis, while alternative splicing of the *LFY*-like gene correlates with flowering ability. Alternative splicing, which occurs in 40–70% of plant genes, significantly increases proteome diversity, and has been documented in the genes involved in flowering and the plant’s response to external cues [36,37,38,39]. In the flowering garlic genotype, *LFY*-like genes are upregulated at least twice, during inflorescence and flower differentiation [13,14]. 

Chromosome-level genome assembly of garlic [8] has provided the opportunity to integrate genome-wide analysis of flowering genes with large transcriptome data. Here, we present detailed genomic and transcriptome analyses of members of the PEBP and *LFY* families in flowering garlic genotypes. 

## 2. Results and Discussion

### 2.1. Discovery of Flowering-Related Genes in the Garlic Genome

Computational analysis of the newly assembled garlic genome [8] for flowering-related genes was performed and compared to large databases of model plants and transcriptome catalogs of bulbous plants, in particular onion and garlic. Numerous homologs of almost all of the main known flowering-related genes were discovered, i.e., floral repressors, photoperiod- and vernalization-related genes, meristem identity, and flower organ differentiation and pollen-development genes, among others (Appendix A). Numerous gene duplications confirmed the general assumption that *Allium* genomes are highly repetitive [8,40,41,42]. Mining of garlic genomic data for the homologs of PEBP and *LFY* families resulted in 29 specific sequences that were used for the integrative analysis with transcriptome catalog (Table 1).

### 2.2. PEBP Genes: Only FT and TFL Clades Are Found in the Garlic Genome 

Previous transcriptome analyses of garlic revealed only seven *FT*-like genes, all defined according to their high homology with onion genes [12,15,16,32]. Genome-wide and phylogenetic analyses revealed a much larger group of PEPB genes—26 garlic homologs of *TFL* and *FT*, but no homologs of MFT of *Arabidopsis, Oryza*, or *Brachypodium* (Table 1, Figure 1A). The MFT clade is perceived as the evolutionary ancestor of the other clades and functions in mosses and gymnosperms. During plant evolution, a first duplication resulted in two families of plant PEBP genes (*MFT*-like and *FT/TFL1*-like), and a second duplication led to the FT/TFL1-like clade that is only present in angiosperms [43,44,45]. *MFT* homologs are upregulated during embryogenesis and highly expressed in mature fruit and seeds, whereas they are not detected in aborted seeds, and markedly decrease during seed germination [46,47]. In some plants, e.g., *Phoenix dactylifera* (date palm) and *Kalanchoë*, *MFT*-like sequences were not identified in the genome data [48,49]. As *MFT*-like genes were not found in the garlic genome or transcriptome, the other PEBP genes might carry out the regulatory functions of this clade.

Whole-genome analysis indicated 6 *TFL1*/*CEN*-like and 20 *FT*-like orthologs while FT orthologs clustered in three subclades (Figure 1A). Similar to other monocots [50], *FT*-like genes in garlic are more diversified than the *TFL1*-like genes. *AsFT1* and *AsFT2* were localized to chromosomes 7 and 6, respectively; both were duplicated (Table 1, Figure 1B) and shared high homology, as well as homology with *AcFT1* and *AcFT2* identified in onion at the transcriptome level [31]. The expression map of PEBP genes in garlic (Figure 1C) showed relatively high expression of *AsFT1.2* in leaves and flowers, whereas *AsFT1.1* was slightly upregulated, and only in flowers. It is possible, therefore, that in garlic, *AsFT1* is involved in the development of flower organs, but not bulbing. Two adjusted paralogs of *AsFT2* also had significantly different expression. *AsFT2.2* was found to be active at several developmental stages: it was upregulated in the meristem after vernalization, and during the growing stage, it was expressed in roots, the basal plate, and especially in the foliage leaves, then in flowers and germinating seeds. Its paralog *AsFT2.1* was only slightly active in flowers and in imbibed and stratified seeds. *FT2* homolog has already been proposed as a florigen in onion and garlic [12,15,31]. Our analysis suggests that in addition to flowering induction, this gene has other functions, for instance, in seed germination (Figure 1C).

**Figure 1 ijms-23-13876-f001:**
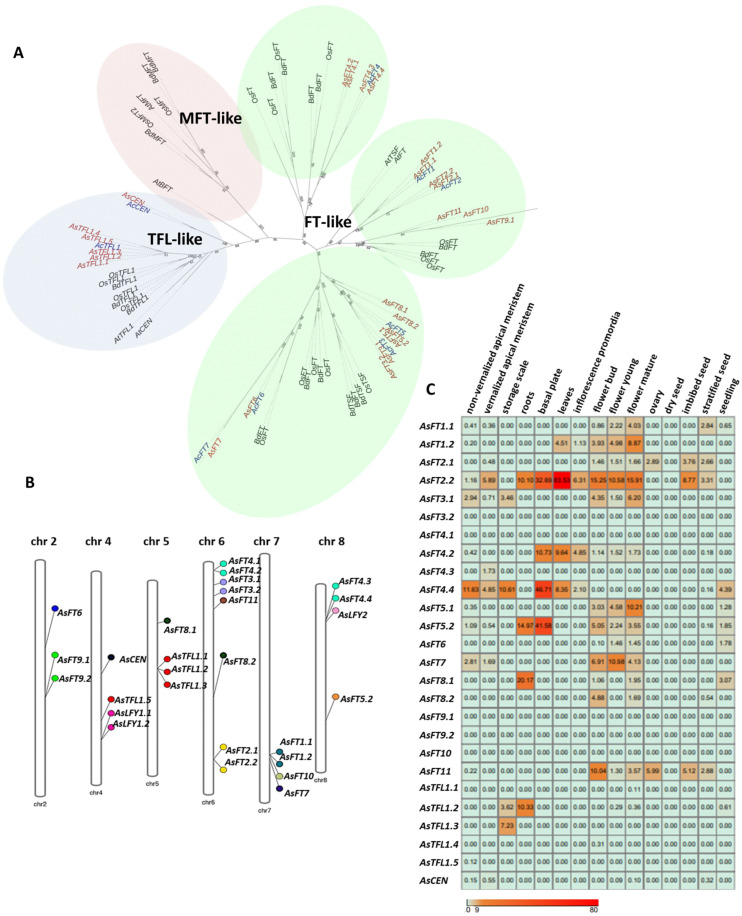
Genomic and transcriptome analyses of PEBP genes in fertile seed-producing garlic genotype. (**A**) Interspecific phylogenetic tree of PEBP proteins from various species: *Arabidopsis* (*Arabidopsis thaliana,* At), onion (*Allium cepa*, Ac), rice (*Oryza sativa*, Os), *Brachypodium* (*Brachypodium distachyon*, Bd), and garlic (*Allium sativum*, As). Alignment was performed using the MAFFT program and RAxML-NG to generate an unrooted ML radial tree constructed with 100 bootstrap replications. iTOL [51] was used for tree presentation. The background colors highlight the groups of FT-like, TFL1/CEN-like, and MFT-like proteins. Red font, garlic genes; blue font, onion genes. (**B**) Chromosomal localization of PEBP and *LFY* genes of *Allium sativum*. The relative sizes represent the length of the garlic chromosomes. Lines connect colored circles to the corresponding base-pair location of each gene. (**C**) Tissue-specific expression patterns of garlic PEBP genes in vegetative and reproductive organs and seeds. The expression values were computed as FPKM based on transcriptome analyses of vegetative and reproductive organs and seeds [11,12,19,52]. FPKM values were used to create a heat map by one minus Pearson correlation, with MORPHEUS [53]. Color scale on heat map ranges from red (high value) to green (low value).

In the same subclade, *AsFT9.1*, *AsFT9.2*, *AsFT10*, and *AsFT11* showed closer homology with monocotyledonous cereal *FT* genes (Figure 1A). These genes were located on different chromosomes (Figure 1B) and their functions are not clear. Only *AsFT11* was upregulated in flowers and germinating seeds, whereas the expression of other genes in this cluster was not recorded in the garlic transcriptome (Figure 1C).

Another subclade of *FT*-like genes included four homologs of onion *AcFT4*—a key player in onion bulbing and bulb maturity [31,32] (Figure 1A). In the garlic genome, two orthologs were localized to chromosome 6 and another pair to chromosome 8 (Figure 1B). Only one copy in each pair was active. *AsFT4.2* was upregulated in the basal plate, leaves, and only slightly in flower buds. *AsFT4.4* was predominantly expressed in vegetative organs and seedlings, but not in the reproductive organs (Figure 1C).

Onion and garlic differ in their physiological responses to environmental signals and mechanisms of flowering and bulbing. In onion, flowering is controlled mainly by temperature, whereas bulbing is induced by long photoperiod. In garlic, however, both flowering and bulbing require lower temperatures, and long days only partly support scape elongation and bulb production [12]. Therefore, the functions of the *FT4* orthologs may differ in these plants. In onion, *AcFT4* is downregulated toward the bulbing stage [31], whereas in garlic, the *AsFT4*s were upregulated during bulbing and are probably involved in its regulation. Moreover, these genes might have other regulatory functions, e.g., involvement in the development of new axillary vegetative buds and subsequent bud/clove dormancy. Since *AsFT4* transcripts are expressed in both vegetative and reproductive organs, these genes might also link bulbing and flowering.

Finally, the third FT subclade contained *AsFT3, AsFT5*, *AsFT6*, *AsFT7* and *AsFT8* (Figure 1A). *AsFT3* and *AsFT5* have high sequence similarity, but they are located on separate chromosomes (Figure 1B). *AsFT3.1* was slightly upregulated in both vegetative and reproductive tissues, whereas *AsFT3.2* remained inactive (Figure 1C). In other species, homologs of *FT3* are involved in both vegetative and reproductive development. Thus, overexpression of barley *HvFT3* promotes flower initiation, but not flower differentiation [54]. In citrus, ectopic expression of *CcFT3* mediates early flowering [55]. In tulips, *TgFT3* is mainly expressed in the leaves and flower stalk of the mother plant prior to the reproductive switch in the axillary buds [28]. However, garlic *AsFT3* does not exhibit any distinct pattern and no specific function is currently attributed to it. At the same time, *AsFT5* paralogs were found to be active in the roots, basal plate, and flowers, and slightly active in seedlings, probably in correlation with the differentiation of axillary meristems. In onion, the expression of *AcFT5* is regulated by photoperiod, but it is not involved in bulb induction [56]. Since *AsFT5.1* and *AsFT5.2* have similar sequences but different expression patterns, they might be two separate genes with different functions.

Two copies of *AsFT8* represented a distinct cluster. They share high homology but are located separately on chromosomes 5 and 6 (Figure 1B) and, similar to the copies of *AsFT5,* have different expression patterns (Figure 1C). Transcriptome analysis revealed high activity of *AsFT8.1* in roots, whereas *AsFT8.2* was slightly upregulated in flowers. Homologs of these genes were not found in the onion transcriptome. 

*AsFT6* and *AsFT7* are clustered together in the phylogram (Figure 1A) but are located on distant chromosomes, and their expression patterns differ. Expression of *AsFT6* was almost undetectable, whereas that of *AsFT7* was upregulated in flowers (Figure 1C). In onion, expression studies of these genes’ homologs have produced contradictory results. Lee et al. [31] suggested that the expression patterns of *AcFT5* and *AcFT6* are not affected by day length, but other reports proposed that *AcFT6* and *AcFT7* are affected by photoperiod [32,56]. In garlic, the functions of these genes need to be further clarified.

In the TFL/CEN-like clade of the phylogram, five *AsTFL1* homologs represented separate clusters (Figure 1A), with high similarity between them (Appendix A). One of these homologs, *AsTFL1.4*, has not been fully sequenced, and we were therefore unable to localize it to a chromosome. *AsCEN* from the same clade was localized to chromosome 4 and has high homology with the recently reported *AcCEN1* (NCBI; accession no. KX257485.1) and *TFL* of *Arabidopsis* (Figure 1A).

The functional role of these genes remains to be elucidated. In model systems, *TFL1* is a key gene for the maintenance of vegetative meristem. TFL1 and FT compete for interaction with chromatin-bound FD, regulating flower induction and shaping inflorescence architecture [23]. However, the functions of *TFL1* homologs differ in plant species [57]. For example, in potato, *StTFL1* acts as a repressor of tuberization [58]. In rice, the anti-florigen *TFL1* is a member of the florigen repression complex (FRC) [59]. In onion, *TFL1/CEN*-like genes are expressed during bulbing and inflorescence development, with a simultaneous reduction in *AcLFY* expression [60]. Our analysis of the garlic transcriptome showed overexpression of *TFL* homologs in roots and storage scales but not in reproductive organs (Figure 1C), suggesting their role in the regulation of bulbing/flowering balance in the flowering garlic genotype. It is not yet clear whether FT and TFL homolog activity is directly linked with the reproductive transition [61], or if they are involved in systemic signaling and coordinating carbon supply at the whole-plant level during induction of flowering [62]. The present analysis suggests that the role of TFL orthologs in garlic florogenesis might be insignificant and that other PEBP genes (of the FT clade) might take over their functions.

### 2.3. Garlic LFY Homologs Are Represented by Multifunctional Genes

Genome-wide analysis of garlic *LFY* homologs revealed three gene sequences. *AsLFY1.1* and *AsLFY1.2* are adjacent on chromosome 4, have 98% similarity, and are highly homologous to onion *AcLFY* (Table 1, Figure 2A). Another gene, *AsLFY2*, is located on chromosome 8 and represents a completely separate line in the phylogram (Figure 2A). The genome sequence of *AsLFY2* (Asa8G01094.1) includes only 166 bp, located between nucleotide positions 318,450,068 and 318,450,234 (Table 1). We analyzed the expression patterns of this sequence in the transcriptome catalog of developing garlic flowers [52] and found that the expression extends both upstream (from 318,449,800) and downstream (to 318,450,400); thus, the actual gene might be longer than 500 bp (Figure 2B).

The expression patterns of *AsLFY*s in garlic organs differ: the two orthologs of *AsLFY1* were upregulated during the floral transition and in inflorescence primordia, whereas *AsLFY2* was expressed in the vernalized apical meristems, differentiating flowers and germinating seeds (Figure 2C). Rotem et al. [14] reported that during garlic florogenesis, *LFY* homologs are upregulated at different developmental stages and might be involved in several processes: meristem transition, inflorescence initiation, and flower organ differentiation. Moreover, RT-PCR expression analysis of the *LFY*-like gene in garlic revealed two splicing variants, long and short—570 bp and 506 bp, respectively. The alternative splicing correlated with flowering ability [13]. Our genome analysis confirms the identity of previously reported transcripts as segments of *AsLFY1* (Appendix A). Furthermore, the spliced segment is delineated as a 64 bp intron in the garlic genome, thus supporting an intron-retention event. The sequence of the garlic intron is 60% identical with those of *A. cepa* (NCBI; accession no. KY985385.1) and *A. fistulosum* (NCBI; accession no. KY985386.1). In both species, it is defined as an intron, suggesting the conserved nature of *LFY*-like sequences in *Allium* species.

In onion, only one *AcLFY* was found at the transcriptome level [56,63], but these data might be updated once the complete onion genome is published. Two *LFY* homologs have been found in several plant species. In oil palm (*Elaeis guineensis*), *OpLFY1* and *OpLFY2* were reported, one of them with a spliced variant (*OpLFY2v*). Similar to garlic, the *OpLFY2v* transcript retains the first intron in the mature mRNA [64]. Both transcripts were expressed only in the apical meristem and in floral tissues. In apple, *AFL1* was expressed only in the floral apex, whereas *AFL2* was also expressed in roots and flower organs [65]. In maize, *ZFL1* and *ZFL2* were expressed in both vegetative apices and reproductive tissue [66].

Alternative splicing in several species also has a distinct impact on flowering time, e.g., *EARLY MATURITY8* (*EAM8*) in barley [67], *FLOWERING CONTROL LOCUS A* (*FCA*) in *Arabidopsis* [68], *FLOWERING LOCUS C* (*FLC*) in *Chrysanthemum morifolium* [69], and *FT* in coconut [70]. Our present analysis implies that in garlic, two *LFY*-like genes independently control different stages of florogenesis, but it is not clear whether both genes act as “pioneer TFs” [35]. 

### 2.4. Gene Co-Expression Network (GCN) in Floral Transition

GCNs demonstrate the integration of multiple datasets by connecting genes with similar expression patterns across treatments or developmental stages. In garlic, several co-expression modules of flowering-related genes reflect interaction of the genes involved in the signal transition and meristem shift to the reproductive stage (Figure 3). PEBP genes play significant roles in the network, with positive and negative correlations between them. Moreover, PEBP genes correlate with and activate numerous genes involved in the floral transition, e.g., floral repressors and genes involved in vernalization and photoperiod pathways, and in meristem maintenance and transition. The GCN representation suggests that most PEBP genes act as regulatory centers for the group of flowering genes. For instance, co-expressed *AsFT2.1* and *AsFT2.2* are positively linked with photoperiodic and vernalization genes. *AsFT8* and *AsFT11*, which are upregulated during early-flower differentiation (Figure 1C), are co-expressed with numerous flowering genes, such as *VIN3*, *LHY*, *CO*, *FLX*, and *FRI* (Figure 3). Therefore, a previous report on the central role of the *FT* family in flower transition in garlic [12,15] is confirmed, while the interplay between these genes occurs both directly and obliquely via other flowering genes.

As has been shown in model plants, the meristem shift requires FT–FD complex formation that, in turn, activates the meristem master regulator *LFY*. During this process, *FT*-like and *TFL1*-like genes act antagonistically in the competition for chromatin-bound FD at shared target loci [33]. Similarly, two orthologs of garlic *AsTFL1* are positively connected, but correlate negatively with the two copies of *AsLFY1*. Although both copies of *AsLFY1* are upregulated during the floral transition, the expression of *AsLFY1.1* correlates with many more genes than that of *AsLFY1.2* (Figure 3). Another *LFY*-like gene, *AsLFY2*, does not show a direct correlation with *AsLFY1*, which suggests an independent role in garlic florogenesis. 

Although orthologs of *AsFT2*, previously proposed to be garlic florigen [15], are co-expressed with numerous genes, the orthologs of *AsFT1* are even more active and directly co-expressed with many flowering genes. At the same time, a negative correlation can be noted between *AsLFY1.1* and *AsFT1.2*, while *AsLFY2* exhibits a direct positive correlation with *AsFT2.1.*

Homologs of *FT4.2* and *FT5.1* clearly interact with numerous flowering genes, including vernalization and photoperiod pathway genes (Figure 3), which are activated by *FT*-like genes in other plants [71,72,73].

The vernalization pathway is known to have evolved a few times in plant evolution and as a result, it is not conserved among plant species [74,75]. A number of the vernalization genes shown in Figure 3, e.g., *FRI*, *VIN3*, and *FLX*, are known to regulate flowering through association with *FLC*, a key flowering repressor in dicotyledons [76,77]. However, *FLC* is not found in garlic, and the functions of homologs of *FRI*, *VIN3*, and *FLX* have yet to be elucidated. 

Two *TFL*-like genes, *AsCEN* and *AsTFL1.4*, are co-expressed mainly with floral repressors and integrators, as well as with many photoperiod genes (Figure 3). In garlic, the initiation of axillary buds occurs concurrently with the floral transition of the apical meristem, and it is therefore possible that one group of PEBP genes regulates the reproductive shift, while another group controls synchronized bulbing and dormancy of the axillary buds, as has been proposed for some geophyte species [18]. Similarly, *TFL1* is overexpressed in dormant buds of walnut, but chilling accumulation downregulates its expression and increases the expression of flowering genes, ultimately leading to dormancy release [78]. In garlic, chilling is also required for both flowering induction in apical meristem and formation of axillary buds [79], and *TFL*-like genes can be involved in dormancy induction and dormancy release in this species. 

## 3. Materials and Methods

### 3.1. Plant Material and Transcriptome Data 

Flowering and seed-setting genotype #87 was selected in 2004 from a segregating seedling population and then clonally propagated at the Agricultural Research Organization (ARO), Volcani Center, Israel. For transcriptome sequencing, tissues were sampled from vegetative and reproductive organs at different stages of plant development and after cold storage treatments (Table 2). The results of the transcriptome analyses were published by our group in 2015–2020 [11,12,19,52,80].

Clean reads from the transcriptomes were mapped to the new reference genome of *Allium sativum* (GCA_014155895.2) [8] using STAR software (v. 2.7.1a) [81], with an average mapping rate of 83.1%. Gene abundance was estimated using Cufflinks [82] (v. 2.2) combined with gene annotations [8]. Integrated gene-expression values were computed as FPKM, and average FPKM values were used to create a combined heat map by one minus Pearson correlation with MORPHEUS [53].

### 3.2. Identification and Sequence Analysis of PEBP and LFY Family Members 

The garlic proteins were used as a query term for a search of the NCBI non-redundant (nr) protein database, carried out with the DIAMOND program [83]. The search results were imported into Blast2GO version 4.0 [84] for gene ontology assignments. Homologous genes were identified by blastx versus *Arabidopsis thaliana* [85], *Brachypodium distachyon* (GCA_000005505), *Oryza sativa* (GCF_001433935), *Allium cepa* (GCA_905187595.1), and *Zea mays* (GCF_902167145) protein sequences with an E-value cutoff of 1e^−5^. Those homology searches were used for genome-wide identification of PEBP and *LFY* genes. *Lilium longiflorum* and *Narcissus tazetta* LFY proteins were extracted from the NCBI database. The related PEBP and *LFY* genes are listed in the Table 1. Garlic genes *AsFT1* to *AsFT6* were named in accordance with the onion *AcFT1* to *AcFT6* genes reported by Lee et al. [31]. *AsFT7* is homologous to *AcFT7* reported by Manoharan et al. [32]. *AsFT8* to *AsFT11* were named according to their distribution in the phylogenetic analysis.

### 3.3. Chromosome Distribution and Gene Structure Analysis

According to the physical locations of each gene on the draft garlic genome [86], the identified *AsFT* genes were mapped onto the corresponding garlic chromosomes using PhenoGram [87]. The detailed positions of these genes are provided in Table 1.

### 3.4. Analysis of AsLFY2 on Chromosome 8

The Integrative Genomics Viewer program [88] was used for manual examination of *AsLFY2* (Asa8G01094) located on chromosome 8 (318,450,068–318,450,234) combined with the reads from the early- and late-flower samples of genotype #87 (using the STAR alignment results).

### 3.5. Phylogenetic Analysis and Multiple Sequence Alignment

Based on previous reports and the homology searches, we collected the FT and LFY proteins of different species [26,30,31,59,60], including *Arabidopsis*, rice, *Brachypodium*, and onion. The sequences of these proteins were downloaded from the NCBI database (https://www.ncbi.nlm.nih.gov/, accessed on 15 February 2021). The detailed sequence accession numbers are listed in Appendix A. Multiple sequence alignments were performed by MAFFT alignment program [89]. A phylogenetic tree was constructed using the RAxML-NG phylogenetic tree tool based on the maximum-likelihood (ML) optimality criterion and the LG empirical substitution matrix [90]; a total of 100 bootstrap replications were used.

### 3.6. Co-Expression Analysis

A gene co-expression network (GCN) was constructed with the Cytoscape software plugin CoNet based on the Pearson correlation. The network was plotted using Cytoscape 3.3.0 [91,92,93].

## 4. Conclusions

Prior to the full sequencing of the extremely large garlic genome in 2020, large transcriptome catalogs were created and used as a basis for garlic genetic studies. Chromosome-level assembly of garlic genome [8] has provided the first opportunity to integrate genome-wide analysis with transcriptome data. We analyzed massive transcriptome data from various growth seasons [5,11,12,19] against the newly published genome for the validation of the transcriptomic results. The genome-wide analysis revealed many PEPB genes that had not been detected in previous transcriptome studies. The newly discovered *FT*-like genes have diverse functions, direct co-expression with flowering genes, and are involved in coordination and balancing of the specific functions of flowering, axillary meristem initiation, dormancy, and seed germination. *MFT*-like genes were not present in the genome, and functions of the *TFL*-like genes were diminished; *FT*-like homologs probably take over their roles. *TFL*-like genes may be involved in maintaining the vegetative status of the apical and axillary meristems and the dormancy process.

The discovery of three sequences of *LFY*-like genes and confirmation of the alternative splicing of *AsLFY1* in the garlic genome refine our previous finding on the key role of these genes in garlic florogenesis. In addition, one *AsLFY* gene is involved in the meristem transition to reproductive development, whereas the second one is expressed during flower differentiation. It is not yet clear whether only *AsLFY1*, or both *AsLFY1* and *AsLFY2*, act as “pioneer TFs”.

The garlic genome is highly repetitive, including key flowering genes. However, orthologs of some flowering genes, e.g., *AsFT2* and *AsLFY1*, differ in their GCN, even if they have similar expression patterns and are found on the same chromosomes. This suggests ongoing evolution in the garlic genome and diversification of gene functions.

Integrative genome and transcriptome analysis of two groups of flowering-related genes in the flowering and seed-producing genotype of garlic confirms the leading role of PEBP and *LFY*-like genes in florogenesis, bulbing, and seed germination in this plant (Figure 4). Orthologs of *AsFT2* and *AsFT4* are upregulated in various organs and developmental stages, and they can be seen as key regulators of vegetative and sexual reproduction. *AsLFY1* and *AsLFY2* differ in their functions, but both are tightly involved in flower initiation and differentiation.

We propose that flowering genes might be conserved in non-bolting, semi-bolting, and bolting genotypes of garlic. However, during crop evolution, the ability to flower and produce seeds was weakened. The process of fertility deprivation might be based on the loss of transcription of the specific flower genes. Further comparison of the genome and transcriptome factors and qRT-PCR validation of PEBP and *LFY*-like genes in various garlic genotypes will strengthen the presented results, clarify the possible evolution of their reproductive traits, and contribute to fertility restoration and efficient breeding of this important crop.

## Figures and Tables

**Figure 2 ijms-23-13876-f002:**
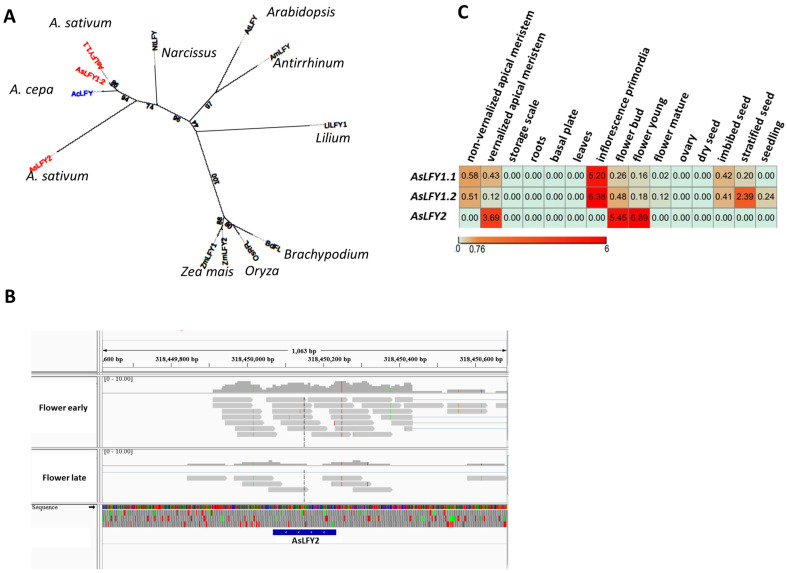
Genome and transcriptome analyses of garlic LFY homologs. (**A**) Interspecific phylogenetic tree of LFY-like protein sequences from onion (*Allium cepa*, Ac), rice (*Oryza sativa*, Os), *Brachypodium distachyon* (Bd), *Narcissus tazetta* (Nt), *Lilium longiflorum* (Ll), corn (*Zea mays*, Zm), *Arabidopsis thaliana* (At), *Antirrhinum majus* (Am), and garlic (*Allium sativum*, As). Alignment performed using the MAFFT program and RAxML-NG to generate an unrooted ML radial tree constructed with 100 bootstrap replications. iTOL [51] was used for tree presentation. Red font, garlic genes; blue font, onion gene. (**B**) The genomic region of *AsLFY2* on garlic chromosome 8. The blue box represents the sequence based on the genome annotation (Asa8G01094.1, Table 1). The gray boxes represent the reads for expression of the transcriptome sequence of the individual flowers [52] upstream and downstream of the genomic sequence of *AsLFY2*. (**C**) Transcriptome analysis of *AsLFY*s in vegetative and reproductive tissues and seeds of garlic [11,12,19,52]. The expression values were computed as FPKM and used to create a heat map by one minus Pearson correlation, using MORPHEUS [53]. Color scale on the heat map ranges from red (high value) to green (low value).

**Figure 3 ijms-23-13876-f003:**
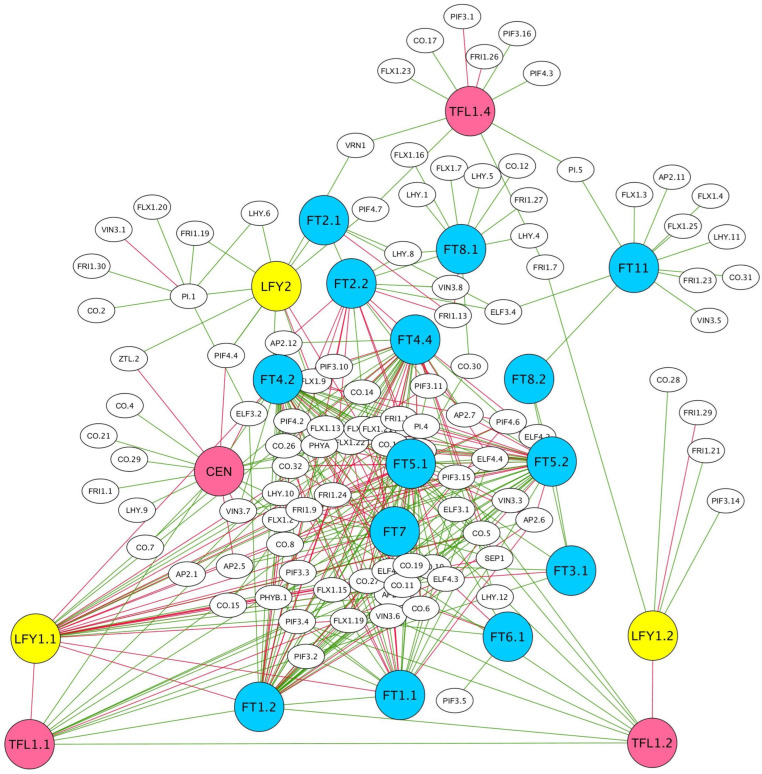
Gene co-expression network (GCN) representation of garlic genes associated with floral transition. Transcriptome data are from the stages of meristem transition and flower differentiation. Transcript sequences were mapped to the garlic genome and analyzed using the network-drawing software Cytoscape. Pearson correlation value higher than 0.75. Positive (green lines) and negative (red lines) correlations between *FT*-like (blue), *TFL1/CEN*-like (pink), *LFY*-like (yellow), and other genes involved in florogenesis are shown.

**Figure 4 ijms-23-13876-f004:**
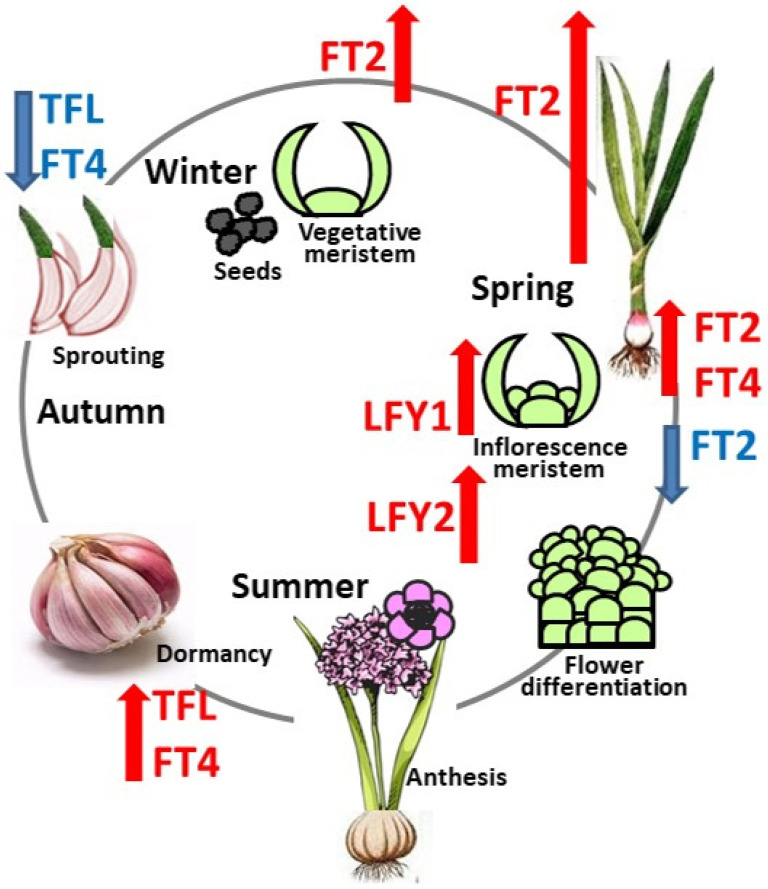
The life cycle of flowering and seed-producing garlic genotype is regulated by PEBP and *LFY*-like genes. Red and blue arrows indicate gene up-regulation and down-regulation, respectively. In summer, dormant meristems express *AsTFL1*, which prevents floral induction, and *AsFT4.4*, which might be involved in bulbing and dormancy. Following winter vernalization, *AsTFL1* and *AsFT4.4* are downregulated, cloves sprout, and during this growth stage, the florigen *AsFT2.2* is upregulated in cloves, basal plate, and leaves. After floral induction, *AsLFY1* acts in the apical meristem and in the developing inflorescence, while *AsLFY2* is only expressed during organ differentiation in the individual flowers. After the floral transition, high expression of *AsFT4.4* in the basal plate supports induction of the developing axillary meristems and bulbing.

**Table 1 ijms-23-13876-t001:** List of flowering-related genes from the PEBP and *LFY* families in the garlic genome, accession IDs from Sun et al. [8].

	Gene Name	Accession ID	Start of Base-Pair Location on Chromosome	End of Base-Pair Location on Chromosome
1	*AsFT1.1*	Asa7G06383.1	1,751,818,141	1,751,820,541
2	*AsFT1.2*	Asa7G06386.1	1,752,347,327	1,752,356,242
3	*AsFT2.1*	Asa6G06199.1	1,723,019,079	1,723,031,572
4	*AsFT2.2*	Asa6G06200.1	1,723,318,919	1,723,326,950
5	*AsFT3.1*	Asa6G00732.1	188,122,544	188,125,123
6	*AsFT3.2*	Asa6G01063.1	285,605,894	285,607,102
7	*AsFT4.1*	Asa6G00187.1	58,999,873	59,000,089
8	*AsFT4.2*	Asa6G00188.1	59,419,215	59,428,386
9	*AsFT4.3*	Asa8G01025.1	303,417,313	303,419,144
10	*AsFT4.4*	Asa8G01036.1	304,772,513	304,774,348
11	*AsFT5.1*	Asa0G05138.1	22,173	24,968
12	*AsFT5.2*	Asa8G04470.1	1,187,736,524	1,187,738,945
13	*AsFT6*	Asa2G02821.1	756,995,398	756,998,250
14	*AsFT7*	Asa7G06501.1	1,791,516,624	1,791,519,654
15	*AsFT8.1*	Asa5G01472.1	361,294,195	361,296,104
16	*AsFT8.2*	Asa6G04367.1	1,172,852,584	1,172,853,324
17	*AsFT9.1*	Asa2G04443.1	1,196,742,904	1,196,743,080
18	*AsFT9.2*	Asa2G04445.1	1,196,975,248	1,196,979,775
19	*AsFT10*	Asa7G06404.1	1,756,659,830	1,756,660,064
20	*AsFT11*	Asa6G01542.1	402,410,306	402,412,331
21	*AsTFL1.1*	Asa5G02970.1	759,662,461	759,663,257
22	*AsTFL1.2*	Asa5G02971.1	759,823,509	759,824,327
23	*AsTFL1.3*	Asa5G02972.1	759,853,529	759,854,342
24	*AsTFL1.4*	Asa0G01615.1	7639	8045
25	*AsTFL1.5*	Asa4G04758.1	1,298,705,414	1,298,706,219
26	*AsCEN*	Asa4G03276.1	890,356,000	890,356,793
27	*AsLFY1.1*	Asa4G05355.1	1,466,526,753	1,466,528,007
28	*AsLFY1.2*	Asa4G05365.1	1,469,543,395	1,469,544,649
29	*AsLFY2*	Asa8G01094.1	318,450,068	318,450,234

**Table 2 ijms-23-13876-t002:** List of samples for transcriptome sequencing, collected at different stages of plant development and after cold-storage treatments. Seed-setting garlic genotype #87 was used in all transcriptome analyses.

Organ/Sample	Description	Bio-Project *	Ref **	Reps ***
Non-vernalized apical meristem	Apical buds from the cloves stored at ambient temperature 20–30 °C	PRJNA566287	[12]	2
Vernalized apical meristem	Apical buds from the cloves stored in at 4 °C for 12 weeks	PRJNA566287	[12]	2
Storage scale	Bulb cloves at the end of the growing season	PRJNA243415	[11]	1
Roots	Fresh roots during active growth	PRJNA243415	[11]	1
Basal plate	Plant basal plate during active growth	PRJNA243415	[11]	1
Leaves	Green foliage leaves during active growth	PRJNA243415	[11]	1
Inflorescence primordia	Young inflorescences with differentiating flower primordia	PRJNA243415	[11]	1
Flower bud	Differentiated flower bud with green tepals, 2.5–3 mm long	PRJNA264944	[52]	3
Flower young	Differentiated flower bud with green tepals, 3–4 mm long	PRJNA264944	[52]	3
Flower mature	Pre-anthesis flower with pink tepals, 3–4 mm long	PRJNA264944	[52]	3
Ovary	Post-fertilization stage, seed setting, ovule length 2–2.5 mm	PRJNA647152	[80]	2
Dry seed	Dry seeds stored after harvest in paper bags at room temperature	PRJNA647152	[19,80]	2
Imbibed seed	Seeds imbibed in water for 6 h	PRJNA647152	[19,80]	2
Stratified seed	Seed stratified at 4 °C for 4 weeks	PRJNA647152	[19,80]	2
Seedling	Seedling with 1–2 leaves	PRJNA647152	[80]	2

* Number in NCBI Sequence Read Archive (SRA) database. ** Reference of the transcriptome analysis. *** Replicates.

## Data Availability

All data generated or analyzed during this study are included in this published article and its Appendix A.

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
