# Peer review of "Integrated Genomic and Transcriptomic Elucidation of Flowering in Garlic"

_ijms, 2022, doi:10.3390/ijms232213876_

Round 1

Reviewer 1 Report

In this manuscript, the authors investigated the genes involved in garlic florogenesis using conjoint genomic and transcriptome analysis. They found that PEBP and LFY genes were not found at the transcriptome level, while FT-like genes replaced the TFL-like genes. Therefore, they proposed that the loss of transcriptional functions in certain specific genes is the main regulation mechanism for garlic fertility deprivation. This is an interesting study, discussing the mechanism regulating garlic fertility deprivation from an evolutionary perspective. The flow of the paper is well organized, and the statements are largely supported by the data. I do not see any need for manuscript content or structure changes, and I recommend speedy acceptance and publication in IJMS.

Author Response

Thank you for the positive feedback

Reviewer 2 Report

Flowering is an important developmental process in plant growth, development, and reproduction. The regulatory mechanism for flowing garlic is still unknown. This study revealed that the transcriptional expressions of phosphatidylethanolamine-binding proteins (PEBP) and LEAFY (LFY) genes were not detectives in garlic, while the role of TFL-like genes was reduced and replaced by FT-like homologs, however, homologs of MFT-like genes were not observed. However, 3 LFY-like genes were identified. 

To sum up, the manuscript was well organized and well written. In addition, the authors present a very interesting topic. There are currently many interests in garlic. It would be of wide interest to the plant community, the crop industry, and the IJMS’s readers.  However, I have some concerns about the manuscript, before publication:

1. The manuscript was based on a large number of transcriptome data, which is very interesting to the readers. To increase the readers’ attention and the citation of the publication. The authors should upload the transcriptome data to a public database, such as NCBI's database.

2. The RNA-Seq data would provide a wide view of the whole transcriptome. And the specific flowering-related gene group needs to be confirmed by qRT-PCR assay. The authors should test their hypothesis by qRT-PCR.

3. The biological replicates are important for readers to understand the data present in the manuscript. The authors need to descript the No. of biological replicates for transcriptome analysis.

4. The authors did not descript clearly how to analyze the transcriptome data. More information in detail needs to be mentioned. For example, how to choose internal control; how to choose parameters to normalize and analyze the gene expression, as well as how they analyze transcriptome data with a “cut-off” value and p-value. The information is important for transcriptome analysis, but the information was missing in the manuscripts.

Author Response

Thank you very much for these comments! I feel that they really helped to improve the manuscript. In the first version, we did not include most of the methods of transcriptome analysis, since we published these methods in our previous papers (2015-2020). Currently, after recent publication of garlic genome, we had a precious option to verify our transcriptomic analyses and to map our transcriptome data on genome.

To clarify the methods of transcriptome analysis in the present manuscript, we included a new Table 2 into section Materials and Methods. This table summarizes all information on tissue samples, data presentation in NCBI, replicate numbers and references for the published reports. I hope that this table actually replies to all comments of the reviewer concerning transcriptome analysis and data interpretation.

As for the proposed validation by qRT-PCR assay, it is certainly an important aspect for our future studies. Such analysis has to be performed at the different stages of plant development, and samples from plants will be collected in vivo.  

Reviewer 3 Report

In this manuscript, the authors take the advantage of the rare flowering and seed-setting garlic genotype 87 to elucidate fertility restoration in garlic cultivars. They have integrated genomic and transcriptomic data to identify genetic determinants of flowering and seed germination and reveal the leading role of PEBP and LFY-like genes. In my opinion, the article is well written, with very interesting results.  

My only remark concerns Fig. 4, which shows a diagram of the role of the studied genes (PEBP and LFY) in flowering and seed-producing processes. The figure shows 3 of the annual seasons, excluding summer. There is also a discrepancy between the diagram and the text to the figure that “Following winter vernalization,  AsTFL1 and AsFT4.4 are downregulated, cloves sprout, and during this growth stage, the florigen AsFT2.2 is upregulated in cloves, basal plate, and leaves” but the designations in the figure do not match this.

Author Response

We would like to thank the reviewer for the comments, his suggestions for the Fig. 4 are correct. This figure was re-designed accordingly

Round 2

Reviewer 2 Report

There are some improvements in the revised manuscript. In addition, I appreciated that the authors answered my concerns one by one. However, I still have some concerns about the revised version:

1. The qRT-PCR will further confirm the hypothesis in the manuscript, which will strengthen the conclusion of this study.

2. The resolution of figure 1, 2 and 3 are not good, please enhance the figures’ resolution. 

Author Response

Thank you again for your comment concerning qRT-PCR. We definitely see this validation as an important part of our research of garlic flowering mechanisms. However, this analysis was not included into the presented manuscript because of two reasons:

  1. Transcriptome analysis includes data from several years. Although the plants of the same genotype were investigated, environmental conditions varied during these years (2014-2021). Therefore, we performed only in silico analysis to open the next stage of the research
  2. We are currently comparing genomic make-up of several garlic genotypes with different flowering ability. qRT-PCR of flowering and non-flowering genotypes are be included into this research.

Therefore, in the presented manuscript we added the following statements:

Prior to the full sequencing of the extremely large garlic genome in 2020, large transcriptome catalogs were created and used as a basis for garlic genetic studies. Chromosome-level assembly of garlic genome [8] has provided the first opportunity to integrate genome-wide analysis with large transcriptome data. We analyzed massive transcriptome data from various growth seasons [5, 11, 12, 19] against newly published genome for the validation of the transcriptomic results.

and: 

Further comparison of the genome and transcriptome factors and qRT-PCR validation of PEBP and LFY-like genes in various garlic genotypes will strengthen the presented results, clarify the possible evolution of their reproductive traits and contribute to fertility restoration and efficient breeding of this important crop.

The additions can be seen in the attached Track Change version of the manuscript. 

As for the figure quality, we attached the separate files of the Figs 1,2,3 in .tiff version, which should be sufficient for the publication

Round 3

Reviewer 2 Report

There are some outstanding improvements in the revised manuscript, as compared to the first version. And I appreciated that the authors answered my concerns and explained the reason why the validation cannot be included in this manuscript. I hope that the resolution of the figures meets the requirement of “IJMS”. To sum up, the manuscript was organized and well written; the analysis and prediction will be interesting to the garlic industry. Therefore, I suggested the manuscript is ready to be published.